# CrossCT: CNN and Transformer cross–teaching for multimodal image cell segmentation

**Sara Joubbi**
Data Science for Health Laboratory (DaScH Lab)
Toscana Life Sciences Foundation
Via Fiorentina 1, 53100 Siena (SI)
`s.joubbi@toscanalifesciences.org`

**Giorgio Ciano**
DaScH Lab
Toscana Life Sciences Foundation
Via Fiorentina 1, 53100 Siena (SI)
`g.ciano@toscanalifesciences.org`

**Dario Cardamone**
DaScH Lab
Toscana Life Sciences Foundation
Via Fiorentina 1, 53100 Siena (SI)
`d.cardamone@toscanalifesciences.org`

**Giuseppe Maccari**
DaScH Lab
Toscana Life Sciences Foundation
Via Fiorentina 1, 53100 Siena (SI)
`g.maccari@toscanalifesciences.org`

**Duccio Medini**
DaScH Lab
Toscana Life Sciences Foundation
Via Fiorentina 1, 53100 Siena (SI)
`d.medini@toscanalifesciences.org`

## Abstract

Segmenting microscopy images is a crucial step for quantitatively analyzing biological imaging data. Classical tools for biological image segmentation need to be adjusted to the cell type and image conditions to get decent results. Another limitation is the lack of high-quality labeled data to train alternative methods like Deep Learning since manual labeling is costly and time-consuming. *Weakly Supervised Cell Segmentation in Multi-modality High-Resolution Microscopy Images*[1] was organized by NeurIPS to solve this problem. The aim of the challenge was to develop a versatile method that can work with high variability, with few labeled images, a lot of unlabeled images, and with no human interaction. We developed CrossCT, a framework based on the cross–teaching between a CNN and a Transformer. The main idea behind this work was to improve the organizers' baseline methods and use both labeled and unlabeled data. Experiments show that our method outperforms the baseline methods based on a supervised learning approach. We achieved an F1 score of 0.5988 for the Transformer and 0.5626 for the CNN respecting the time limits imposed for inference. The code is available on GitHub `https://github.com/dasch-lab/crossct`.

## 1   Introduction

Microscopy image segmentation is often a crucial step in the quantitative analysis of imaging data for biological applications [1]. Usually, the identification of nuclei via segmentation is the first step to detect single cells in an image and perform subsequent tasks, e.g. counting cells [2], tracking moving populations [3], and subcellular localization of protein signal  [4].

---

[1] `https://neurips22-cellseg.grand-challenge.org/`

36th Conference on Neural Information Processing Systems (NeurIPS 2022).

Most of the existing bioimage analysis tools identify nuclei using classical segmentation algorithms.These methods commonly consist of sophisticated combinations of pre-processing filters, e.g., Gaussian or median filters, and segmentation operations, e.g., a region adaptive thresholding followed by a watershed transformation [5]. The main problem with these algorithms is that traditional methods need to be adjusted to the cell type and image conditions. However, a controlled experimental setting is not sufficient to find a unique choice of parameters that can correctly segment all the images. In fact, classical algorithms can fail to adapt to the heterogeneity of biological samples or can be sensitive to technical artifacts.

Deep Learning (DL) algorithms have shown encouraging results in fully supervised image segmentation [6, 7], outperforming traditional methods even on very diverse datasets. To achieve good performance and improve the generalization ability, DL models require a diverse and large amount of high-quality labeled data. However, creating datasets with these requirements is extremely laborious and time-consuming. Such an issue is more noticeable in the field of microscope imaging where the resolution is high. Transfer learning was proposed to address the scarcity of data by transferring the data distribution from the source domain to the target domain [8, 9, 10]. Another way to address data scarcity is to apply weakly-supervised and semi-supervised learning. In recent years, numerous weakly-supervised segmentation techniques have been developed, the main idea is to use as less annotation as possible (e.g. image-level labels [11, 12], bounding box annotation [13, 14], and point annotation [15, 16, 17, 18]). On the other hand, semi-supervised learning aims to construct models that use both labeled and unlabeled images (e.g. consistency regularization [19, 20], GAN-based approach ( [21, 22]).

The 'Weakly Supervised Cell Segmentation in Multi-modality High-Resolution Microscopy Images' competition was organized by Neural Information Processing Systems (NeurIPS) to challenge the participants to find cell segmentation methods that could be applied to various microscopy images across multiple imaging platforms and tissue types. The goal is to create a generic, reusable model that is trained once and can be reused on various microscopy experiments without further user intervention. The task's difficulty is working with an extremely variable dataset, both in terms of the type and size of the cell, and in terms of acquisition techniques. In addition, the dataset has limited labeled images and many unlabeled images (unlabeled images are relatively easy to obtain in practice).

In this paper, we present CrossCT, a framework based on cross-teaching between a Convolutional Neural Network (CNN) and a Transformer. Our method benefits from the two different learning paradigms: CNN is inadequate in learning global context and long-range spatial relations; transformers can capture long-range feature dependencies, but the lack of low-level details may result in limited localization capabilities. For instance, CNN-based deep networks generally have weak performances, especially when target structures exhibit significant variation in texture, shape, and size. Hence, long-range dependency learning could help to prevent the segmentation network from making this mistake. This paradigm of learning global and local features has proven effective in object detection [23, 24] and image segmentation [25].

The paper is organized as follows. In Section 2, the proposed method is presented. Section 3 describes the dataset and the training protocol. Section 4 shows and discusses the experimental results. Finally, in Section 5, we draw the conclusions.

## 2   Method

Our method is represented in Fig 3. The framework is composed of two networks: a CNN (U-Net) that learns the local features in the images, and a Transformer (Swin Transformer + U-Net) that learns the global features in the images. The cross-teaching method is based on a previous work [25] that applies a similar framework to biomedical images. The main differences between our work and [25] are the following. Firstly, the original work directly uses instance labels, while our network works with two different types of labels. In particular, background, interior and boundary are used as classes, but we also exploited two distance maps, neighbor maps and distance maps from the cell center. Secondly, we designed a new loss that takes into account the two different kinds of labels. The proposed scheme implicitly encourages consistency between the two networks, combining the advantages of CNNs and Transformers to compensate each other and resulting in better performance.

In addition, this approach uses both labeled and unlabeled images, contrary to the baseline methods provided by the organizers that used just a supervised approach.

## 2.1 Preprocessing

The dataset provided by the NeurIPS challenge organizers has been generated with four different acquisition techniques (Brightfield, Fluorescent, Phase-contrast, and Differential interference contrast), and therefore the images presents a different number of channels. To uniform the dataset format, all images were converted to three channels, and the channels were repeating two times for one-channel images. Then, the images were processed starting with an intensity normalization provided by the organizers, which makes the nuclei more visible to the network (Fig 1). Finally, the images were saved using a uint16 format.

As for the labels, instead of using the instance representation, the organizers proposed a three-class representation (background, interior, and boundary) to help the network separating the nuclei. To further improve cell boundary recognition, we added two more information: the distance from the center of the cell and the neighbor distance between cells. Cell distances are generated from ground truth data by computing the Euclidean distance transform for each cell independently, while the neighbor distances are computed considering each pixel of a cell as the inverse normalized distance to the nearest pixel of the closest neighboring cell. This representation was defined in [26] to solve the challenging problem of segmenting touching cells of various types in the absence of large training datasets. The three-class labels were saved in an uint16 format and the two distance maps were saved in a float32 format. In Fig 2 the five classes used for the training are shown.

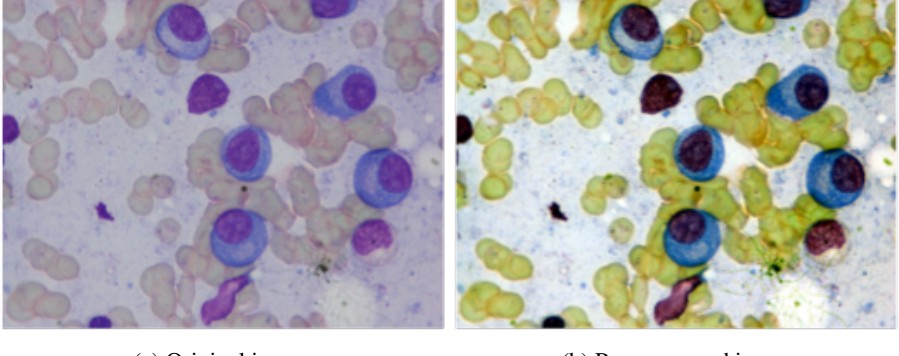

(a) Original image                              (b) Pre-processed image

Figure 1: Intensity normalization applied to the images.

## 2.2 NeurIPS baseline models

NeurIPS provided three different models as baselines for the challenge [2] : U-Net [27], ViT + U-Net [28], and Swin Transformer + U-Net [29]. Adam optimizer [30] with an initial learning rate of 6e-4 was used for the training. The batch size was set to 8 and the maximum number of epochs to 2000 with an epoch tolerance of 100. The dataset used was the labeled dataset provided by the organizers (training: 900 images, validation: 100 images). The code is implemented using PyTorch [31] and MONAI library [3]. Table 4 shows the results of the three models' training. The Swin Transformer + Unet is the better-performing model, followed by the U-Net.

## 2.3 CrossCT

The proposed solution is based on the Cross Teaching between a CNN and a Transformer network. The CNN component is the U-Net provided by the organizers using the MONAI framework. The input to the network is a patch of 3 channels×256 pixels×256 pixels obtained from the original

---

[2] https://neurips22-cellseg.grand-challenge.org/baseline-and-tutorial/
[3] https://monai.io/index.html

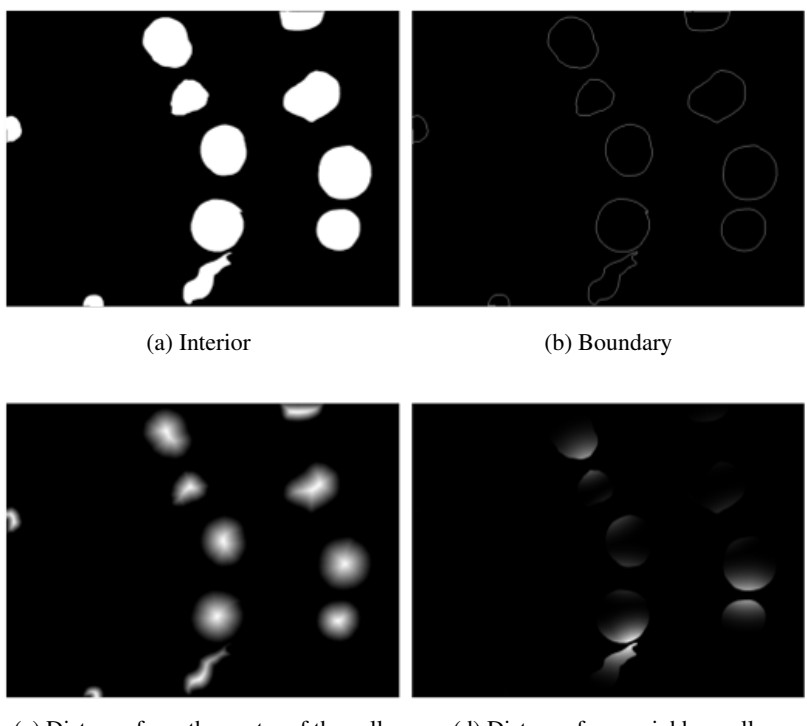

(a) Interior                      (b) Boundary

(c) Distance from the center of the cell     (d) Distance from neighbor cells

Figure 2: Different representations used for the training.

training images. The output is the same patch with the 5 classes representation. The U-Net is a
5-layer network with down/upsampling by a factor of 2 at each layer with 2 convolution residual units.
The transformer component of the framework is the U-Net architecture with a Swin transformer
encoder. Even in this case, we used the baseline model developed using MONAI. The network has a
3 channels patch input with a size (of 256,256), 5-channel output, and a feature size of 24.

The loss function combines of the supervised loss and the cross-teaching loss. In each loss, we have
considered the classification task (detection of background, interior, and boundary) and the regression
task (distance maps and neighbor maps). The classification task was performed using the summation
between Dice loss [32] and cross-entropy loss because compound loss functions have been proven to
be robust in various medical image segmentation tasks [33]. The regression problem was carried out
using the Mean Squared Error (MSE) loss function used on [26].

For the labeled data, the CNN and transformer are supervised by the ground truth individually. For
an input image $x_i$, the proposed framework produces two predictions:

$$\mathrm{p}_i^c = \mathrm{f}_\phi^c\left(\mathrm{x}_i\right); \quad (1) \qquad \mathrm{p}_i^t = \mathrm{f}_\phi^t\left(\mathrm{x}_i\right) \quad (2)$$

where $p_i^c$, $p_i^t$ represent the prediction of a CNN ($f_\phi^c(.)$) and a Transformer ($f_\phi^t(.)$), respectively. Each
network predicts the 3 class labels ($p_{i,3c}^c$ for the CNN and $p_{i,3c}^t$ for the Transformer) and the distance
and neighbor maps ($p_{i,dn}^c$ for the CNN and $p_{i,dn}^t$ for the Transformer). Considering $y_{i,3c}$ and $y_{i,dc}$
the ground truth labels for the 3 classes and the distance and neighbor maps, the supervised loss is
computed as the sum of the two networks' supervised loss:

$$\mathcal{L}_{sup} = \mathcal{L}_{sup_1} + \mathcal{L}_{sup_2} \qquad (3)$$

where,

$$\mathcal{L}_{sup_1} = DiceCE(\mathrm{p}_{i,3c}^{c}, \mathrm{y}_{i,3c}) + MSE(\mathrm{p}_{i,dn}^{c}, \mathrm{y}_{i,dn}) \tag{4}$$

$$\mathcal{L}_{sup_2} = DiceCE(\mathrm{p}_{i,3c}^{t}, \mathrm{y}_{i,3c}) + MSE(\mathrm{p}_{i,dn}^{t}, \mathrm{y}_{i,dn}) \tag{5}$$

are the loss computed for each network as the sum of the DiceCE Loss of the 3 class labels and the MSE of the distance and neighbor maps.

Then the predictions of unlabeled images generated by CNN/Transformer are used to update the parameters of the Transformer/CNN respectively. Based on the predictions of $(f_\phi^c(.))$ and $(f_\phi^t(.))$, the pseudo labels for the 3 classes for the cross teaching strategy are generated by this way:

$$\mathrm{pl}_{i,3c}^{c} = argmax\left(\mathrm{p}_{i,3c}^{c}\right) = argmax\left(\mathrm{f}_\phi^c\left(\mathrm{x}_i\right)\right) \tag{6}$$

$$\mathrm{pl}_{i,3c}^{t} = argmax\left(\mathrm{p}_{i,3c}^{t}\right) = argmax\left(\mathrm{f}_\phi^t\left(\mathrm{x}_i\right)\right) \tag{7}$$

The cross-teaching loss is computed as the sum of the DiceCE Loss between the prediction of CNN and the pseudo label of the Transformer and vice versa. Then we sum the MSE between the prediction of the CNN and the prediction of the Transformer, as in the following equations:

$$\mathcal{L}_{ctl} = \mathcal{L}_{ctl_1} + \mathcal{L}_{ctl_2} + \mathcal{L}_{regression} \tag{8}$$

where,

$$\mathcal{L}_{ctl_1} = DiceCE(\mathrm{p}_{i,3c}^{c}, \mathrm{pl}_{i,3c}^{t}), \tag{9}$$

$$\mathcal{L}_{ctl_2} = DiceCE(\mathrm{p}_{i,3c}^{t}, \mathrm{pl}_{i,3c}^{c}) \tag{10}$$

$$\mathcal{L}_{regression} = MSE(\mathrm{pl}_{i,dn}^{c}, \mathrm{pl}_{i,dn}^{t}). \tag{11}$$

The final loss is the sum of the supervised loss and a weight factor multiplied by the cross-teaching loss function:

$$\mathcal{L}_{total} = \mathcal{L}_{sup} + \lambda\mathcal{L}_{ctl} \tag{12}$$

The weight factor is defined by a time-dependent Gaussian warming-up function commonly $\lambda(t) = 0.1 \cdot e^{-5(1-\frac{t_i}{t_{total}})^2}$.

## 2.4 Post-processing

Regarding post-processing, we used the one provided by the organizers, which defines the instances starting from the interior map predicted by the network.

# 3 Experiments

## 3.1 Dataset

The training set provided by the organizers consists of 1000 labelled images and 1726 unlabeled images originating from various microscopy types, tissue types, and staining types. There are four microscopy modalities in the training set, including Bright Field (BF), Fluorescent (fluor), Phase-Contrast (PC), and Differential Interference Contrast (DIC). The dataset has different types of cells, e.g. bone marrow, primary dermal human fibroblast cells, induced leukocyte stem cells, platelets, and saccharomyces cerevisiae cells. Moreover, the images have different features in terms of the number of nuclei per image and different cell dimensions. The validation set comprises 100 unlabeled images and the test set of more than 200 images.

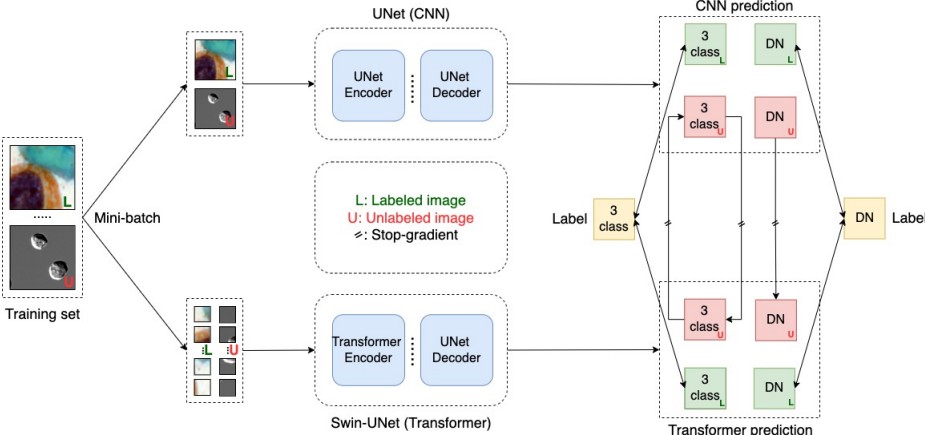

Figure 3: Cross–teaching between CNN and Transformer. Labeled and unlabeled images pass through both networks. The two networks predict the 3 classes (background, interior, and boundary) and Distance and Neighbor maps (DN). For the supervised branch, the predictions are compared with the ground–truth labels. For the cross–teaching branch, the prediction of the 3 classes of one network is compared with the pseudo-label of the other, and the prediction of the DN labels of the two networks is compared with each other.

Two additional public datasets were included in the original one (Cellpose [34] and Omnipose [35]) with the aim of helping the network generalizing more widely and more robustly. The Cellpose dataset is composed by 608 highly-varied images of cells, containing over 70,000 segmented objects. Those images contain different types of cells (e.g. neurons, macrophages, epithelial and muscle cells, as well as plant cells), a small set of microscopy images that did not contain cells or contained cells from very different types of experiments, and a small set of non-microscopy images (e.g. fruit, rocks, and jellyfish). The Ominpose dataset has 735 bacteria images originating from four different sources using distinct microscopes, objectives, sensors, illumination sources, and acquisition settings.

## 3.2 Implementation details

The code is implemented using PyTorch [31] and MONAI library [4]. MONAI is an open-source framework that is built on top of PyTorch. More information about the environment are shown in Table 1.

The final evaluation of the model is performed using two metrics: F1 score and time efficiency. The F1-score is computed at the IoU threshold 0.5 for the true positive. The time efficiency shown in the equation 13, considers the prediction time and the time tolerance for the docker startup time. Specifically, the time tolerance is 10s if the image size (height H x width W) is no more than 1,000,000. If the image size is more than 1,000,000, the time tolerance is (HxW)/1000000x10s.

$$\text{Time Tolerance(H, W)} = \begin{cases} 10\text{s}, & \text{if } H \times W \leq 10^6 \\ \frac{H \times W}{10^6} \times 10s, & \text{if } H \times W > 10^6 \end{cases},$$

$$\text{Running time} = max(0, T - \text{Time Tolerance})$$

(13)

### 3.2.1 Environment settings

The development environments and requirements are presented in Table 1.

### 3.2.2 Training protocols

Our network architecture consists of a U-Net and a Swin transformer U-Net provided as the baseline by the organizers, as already described in the previous section, but with some training modifications Table 2. A final dataset of 2343 labelled images (NeurIPS dataset, Cellpose dataset, and Omnipose

---

[4]`https://monai.io/index.html`

Table 1: Development environments and requirements.

| | |
|---|---|
| System | Ubuntu 20.04.5 LTS |
| CPU | AMD EPYC 7413 24-Core Processor |
| RAM | 16×4GB; 2.67MT/s |
| GPU (number and type) | NVIDIA A100-SXM-80GB |
| CUDA version | 11.5 |
| Programming language | Python 3.8.13 |
| Deep learning framework | Pytorch [31] (Torch 1.12.1, torchvision 0.13.1) |
| Code | `https://github.com/dasch-lab/crossct` |

dataset) was used to train the two baseline models. The dataset was splitted in a training set (70%) and as a validation set (30%) for the performance assessment. Data augmentation was applied to the training images, following the same procedure provided by the organizers. Image size was uniformed by randomly sampling patches of 256×256 from the original dataset, and we used a sliding window of 256×256 for the inference. During the training, we evaluate the validation dataset and we saved the model that had a higher F1 score.

Once the baseline was trained, we used the U-Net and the Swin Transformer + U-Net model as the pre-trained model for the cross-teaching between those two networks. This makes cross-teaching faster than starting from scratch. The cross-teaching protocol is defined in Table 3 and we have followed the same procedure illustrated for the baseline training, but we used just the NeurIPS training dataset (1000 labeled images and 1726 unlabeled dataset).

In Table 4 we have also included a cross-teaching between ResNet + U-Net [36] and Swin + U-Net to evaluate if a different network could achieve better results with respect to the U-Net. The ResNet + UNet was modified with 5 layers as the U-Net, and the number of parameters is twice the U-Net (ResNet + U-Net: 3.23M and U-Net: 1.63M 3). We directly pre-trained the ResNet + U-Net as the other networks and use it for cross-teaching.

Table 2: Training protocols for the two baseline models: U-net and Swin transformer + U-net.

| | |
|---|---|
| Network initialization | "he" normal initialization |
| Batch size | 64 |
| Patch size | 256×256 |
| Total epochs | 2000 |
| Optimizer | Adam |
| Initial learning rate (lr) | 0.1 |
| Lr decay schedule | - |
| Training time | 1 week |
| Loss function | Dice Cross Entropy + Mean Squared Error |
| Number of model parameters | U-Net: 1.63M ; Swin Transf + U-Net: 6.29M[5] |
| Number of flops | U-Net: 1.27G; Swin Transf + U-Net: 4.87G[6] |
| $CO_2$eq (Optional) | - |

## 4   Results and discussion

The cross-teaching method exploits labeled and unlabeled images, where the unlabeled data prediction is used as the pseudo label to directly supervise the other network end-to-end. The strategy of cross-teaching can produce more stable and accurate pseudo labels than explicit consistency regularization. Hence, the use of unlabeled images has improved the performance of the two networks compared to the two baselines, as shown in Table 4.

During the training and evaluation phase, the Swin Transformer U-Net (F1 score: 0.5988) performed better than the U-Net (F1 score: 0.5626). Although the first network has a higher score, we chose the second one because it is faster in performing the segmentation, since the prediction time is part of the evaluation. The tuning set analysis highlighted the efficacy of U-Net in recognizing nuclei of different dimensions and shapes but still does not separate the nuclei properly when the cells are

Table 3: Training protocols for the cross-teaching between the U-Net and the Swin transformer + U-Net.

| Network initialization | pre-trained baseline models (Table 2) |
|---|---|
| Batch size | 64 |
| Patch size | 256×256 |
| Total epochs | 50,000 |
| Optimizer | Adam |
| Initial learning rate (lr) | 0.01 |
| Lr decay schedule | - |
| Training time | 4 weeks |
| Loss function | Dice Cross Entropy + Mean Squared Error |
| Number of model parameters | U-Net: 1.63M ; Swin Transf + U-Net: 6.29M[7] |
| Number of flops | U-Net: 1.27G; Swin Transf + U-Net: 4.87G[8] |
| $CO_2$eq (Optional) | - |

close or merged together. One of the reasons could be that the boundary was not correctly detected during the prediction and this requires more accurate post-processing to define them better.

## 4.1 Quantitative results on tuning set

The F1 score obtained on the tuning set for the different models is presented in Table 4. The unlabeled images improved the performance of the baseline models. Since we have introduced unlabelled images for the training, we need to check if CrossCT has better performance with the whole dataset. As reference methods for the ablation study, we used the U-Net and Swin + U-Net pre-trained with fully-supervised learning. Unfortunately, the tuning set labels were not available at the time of writing, hence we have decided to compare both models with a subset of the labeled dataset (our validation dataset used during the training). Figure 4 shows the comparison between the different networks in a bone marrow image. The cross-teaching and the addition of the unlabelled data allowed the CrossCT method to separate the cells better and to have a more "clear" output than the U-Net trained just with a fully supervised approach. Additionally, table 4 shows that the Swin + U-Net during the cross-teaching is performing better and learns faster than the U-Net only if the network is pre-trained. This difference in performance could be explained by the already demonstrated capacity of transformers to perform better than CNNs when it comes to transferring knowledge as demonstrated in [37].

## 4.2 Qualitative results on validation set

Figure 5 shows some qualitative results of the CrossCT U-Net for different type cells in the tuning set. Images with fully separated cells are properly segmented regardless of cell type and shape. In Fig. 5(a) the violet cells are correctly segmented by the network. Moreover, the platelet images are correctly segmented even if the boundary of the cells is difficult to define starting from the image. Generally speaking, the network fails to correctly separate cells that are near or merged together and with too high or too low nuclei dimensions, even if the network is able to detect the cells (e.g., in fluorescence images). This could be solved with better post-processing that combines the 5 different outputs of the network. Fig. 5(b) highlights issues in segmenting specific types of images (e.g., bone marrow and fluorescence). A possible cause could be the relatively poor representation of those types of images in the training dataset. This problem could be solved using a more balanced dataset and more data augmentation for the different cells.

## 4.3 Segmentation efficiency results on validation set

The segmentation efficiency for our network is represented in Table 5. The overall ranking time is 3.0778 seconds, dividing this time by the 101 validation images, we obtain a mean time of 0.03 seconds.

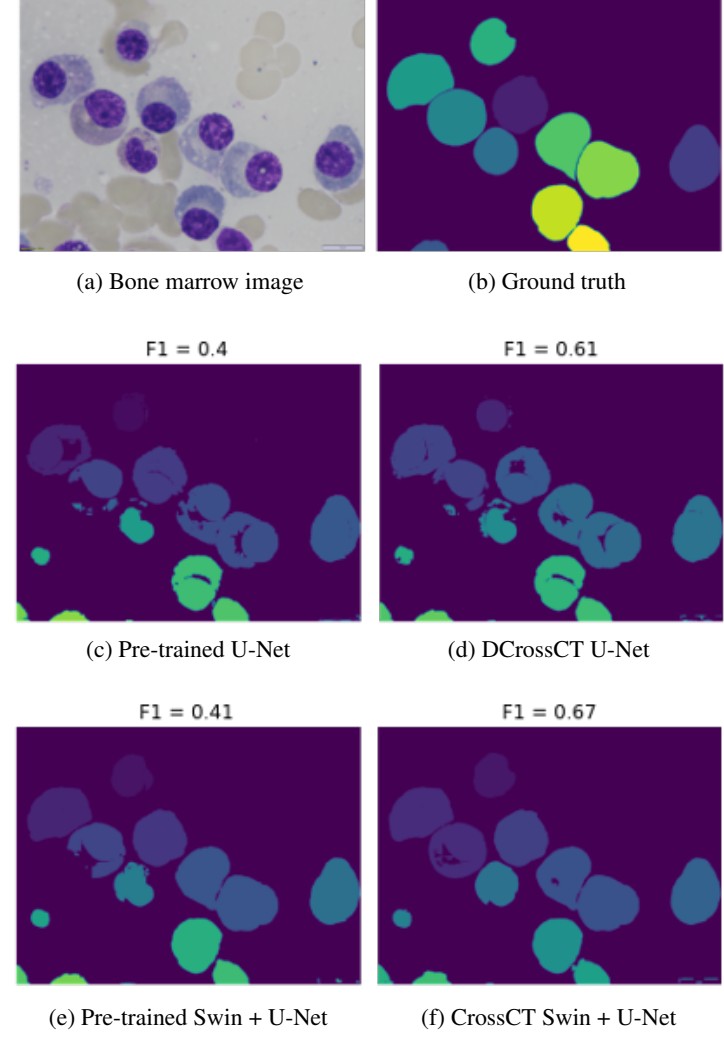

(a) Bone marrow image      (b) Ground truth

F1 = 0.4      F1 = 0.61

(c) Pre-trained U-Net      (d) DCrossCT U-Net

F1 = 0.41      F1 = 0.67

(e) Pre-trained Swin + U-Net      (f) CrossCT Swin + U-Net

Figure 4: Comparison between CrossCT and the pre-trained model on bone marrow images.

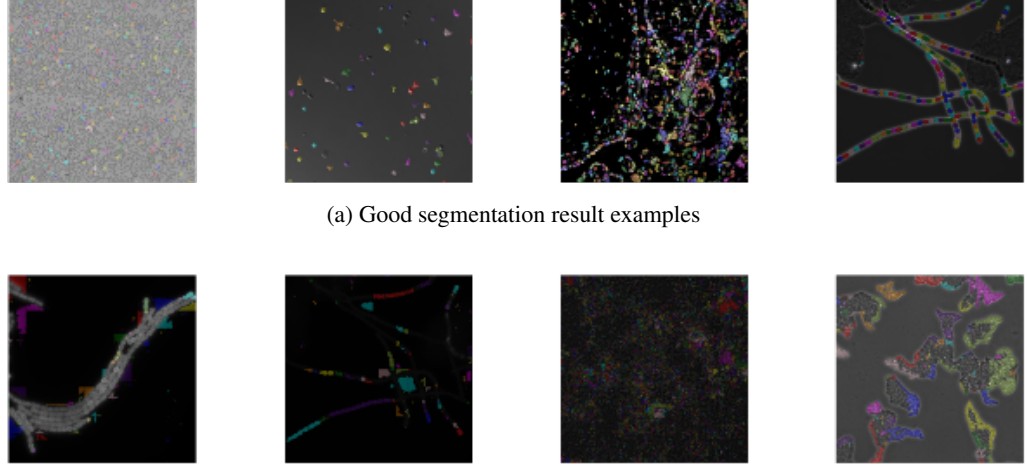

(a) Good segmentation result examples

(b) Poor segmentation examples

Figure 5: CrossCT U-Net prediction with different types of cells.

Table 4: F1 score evaluation for the different models used in this study.

| Phase | Classes | Learning | Additional dataset | Epochs | Model | Best mean dice (training) | Mean F1 score (training) | Mean F1 score (submission) |
|---|---|---|---|---|---|---|---|---|
| baseline | 3 | supervised | - | 2,000 | U-Net | 0.7119 | 0.6463 | 0.4937 |
| | | | | | VIT + U-Net | 0.5915 | 0.2790 | 0.2828 |
| | | | | | Swin + U-Net | 0.7286 | 0.6735 | **0.5482** |
| | | | | | ResUnet | 0.6721 | 0.6241 | 0.5466 |
| pre-trained | 5 | supervised | Cellpose Omnipose | 2,000 | U-Net | 0.7970 | 0.5733 | 0.5335 |
| | | | | | Swin + U-Net | 0.8089 | 0.6301 | **0.6015** |
| | | | | | ResUnet | 0.8023 | 0.6287 | 0.6011 |
| cross-teaching | 3 | semi-sup | - | 2,000 | U-Net | 0.6191 | 0.3158 | 0.2225 |
| | | | | | Swin + U-Net | 0.5909 | 0.2102 | 0.2185 |
| | | | | 50,000 | U-Net | 0.7062 | 0.5339 | **0.5339** |
| | | | | | Swin + U-Net | 0.7038 | 0.4488 | 0.4437 |
| CrossCT | 5 | semi-sup | - | 2,000 | U-Net | 0.5939 | 0.3821 | 0.3369 |
| | | | | | Swin + U-Net | 0.6204 | 0.4485 | 0.4448 |
| | | | | 50,000 | U-Net | 0.7280 | 0.6896 | 0.5626 |
| | | | | | Swin + U-Net | 0.7360 | 0.7068 | **0.5988** |
| CrossCT | 5 | semi-sup | - | 2,000 | ResUnet | 0.6961 | 0.6229 | 0.5387 |
| | | | | | Swin + U-Net | 0.6236 | 0.5062 | 0.4332 |
| | | | | 50,000 | ResUnet | 0.7133 | 0.6835 | 0.5700 |
| | | | | | Swin + U-Net | 0.7383 | 0.6950 | **0.6059** |

Table 5: Running time evaluation.

| Img Name | Real Running Time (s) | Rank Running Time (s) |
|---|---|---|
| cell_00001.tiff | 13.0778 | 3.0778 |
| from cell_00002.png to cell_00100.tif | average: 8.0072 | 0.0 |
| cell_00101.tif | 28.5264 | 0.0 |

## 4.4 Results on final testing set

The final ranking of the challenge was made by evaluating the model on the test set. Table 6 shows the F1 score for different types of images. CrossCT achieves interesting results on all types of images, except fluorescence images. This is probably because fluorescence images have a higher number of nuclei, which are very dense and of different shapes. In fact, in the fluorescence labels, there is no clear edge between the different cells. This aspect could be solved by using the distance and neighbor maps, which will also improve the prediction of the whole dataset.

Table 6: Results on the test set

| Median FI-AlI | Median FI-BF | Median FI-DIC | Median FI-Fluo | Median FI-PC | Mean FI-AlI | Mean FI-BF | Mean FI -DIC | Mean FI-Fluo | Mean FI-PC |
|---|---|---|---|---|---|---|---|---|---|
| 0.3463 | 0.4401 | 0.4005 | 0.0088 | 0.5268 | 0.3437 | 0.4408 | 0.3856 | 0.0878 | 0.4816 |

## 4.5 Limitation and future work

Dataset image variability is the main limitation, some categories are more numerous than others (e.g. bone marrow and fluorescence images have a high number of examples). The dataset imbalance makes it more difficult for the network to learn segmenting different types of cells. Moreover, the difference in cell sizes within the image increases the difficulty of the segmentation task. Another limitation is that the networks still can not perfectly separate all nuclei. Hence, one of the main future development is to develop a more sophisticated post-processing combining the prediction of the three classes with the distance and neighbor maps to improve cell separation.

Next step would be to analyze the performance of our framework with a semi-supervised method to see if we have achieved similar or better results. Moreover, the main idea of the paper was to improve the baseline method and use the unlabelled images. However, giving the good performance of the

ResUnet, testing different combinations of networks could achieve better results. We should also exploit different semi-supervised learning techniques to analyze which one is the best in transferring knowledge.

An interesting aspect to further analyse is the error propagation using pseudo labels that can also lead to low performance. In self-labelling network, the detector is misguided by the incorrect pseudo labels predicted by itself (dubbed self-errors). Teacher-student network are not enough to solve this problems since pseudo labels always remain fixed, and the teacher detector do converge to the student detector in the late stage of training, thus the labeling process degenerates into the self-labeling manner and suffers from the same limitations. Cross Pseudo Supervision is a good method to limit this errors [38]. This work consists of two parallel segmentation networks that have the same structure and their weights are initialized differently. This could be more similar to our case. However, [39] showed that the cross-pseudo supervision methods cannot fully exploit the advantages of multiple models and improve the quality of pseudo labels. The authors proposed a framework that leverages the disagreements between networks to discern the self-errors and refines the pseudo label quality by the proposed cross-rectifying mechanism. Hence, future development could include also a more detailed study on error propagation with this different semi-supervised learning approaches.

# 5    Conclusions

In conclusion, in this work, we present CrossCT, a generic and reusable model based on cross-teaching between CNN and Transformer , able to segment a variety of different microscopy experiments, without additional user intervention. The idea is based on the assumption that CNN can capture local features efficiently and Transformer can model the long-range relation better, and these properties can complement each other during training. Experimental results showed that the proposed method can outperform the supervised method provided by the organizers as a baseline. In the future, more sophisticated pre-processing techniques will be implemented to resize the nuclei dimension to improve the high dimension nuclei detection and separation. In addition, more efficient post-processing technique will be adopted to better separate the nuclei. A starting point could be to combine the 3 classes with the distance labels and the neighbor labels.

# Acknowledgement

The authors of this paper declare that the segmentation method they implemented for participation in the NeurIPS 2022 Cell Segmentation challenge has not used any private datasets other than those provided by the organizers and the official external datasets and pretrained models. The proposed solution is fully automatic without any manual intervention.

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
