# OpenReview forum: "CrossCT: CNN and Transformer cross–teaching for multimodal image cell segmentation"
_NeurIPS.cc/2022/Challenge/CellSeg — Submitted to NeurIPS CellSeg 2022_

### Official Review · Reviewer_jKMD · 2022-12-18
**Cross-teaching as a way to use labelled and unlabelled data for bio-medical image segmentation**

**Rating:** 9
**Confidence:** 5

**Review:**

Summary
========

This paper proposes a framework based on cross-teaching between a CNN and a Transformer to exploit both labeled and unlabeled images as an alternative to fully supervised learning approaches. The clear advantage of such an approach is that it does not requires a lot of training data coming from a single imaging modality, can generalize well on images coming from varied imaging modalities and benefits from CNN and Transformer architectures to create a pseudo-label for the unlabeled images as a part of the training process. The network has a supervised branch (for labeled images) and a cross-teaching branch (for unlabeled images), the loss terms compare the pseudo-labels generated by the CNN and Transformer. The network also has regression loss terms to produce the distance and neighbor labels along with predicting background, interior and boundary pixels.

Training Methodology
-----------------------------
The authors train the CNN (U-Net) and the Transformer (Swin Transformer + U-Net) individually which are then used as pre-trained models during the cross-teaching between the two networks. During the base model training a combination of NeurIPS provided data with Cellpose and Omnipose data was used for model trainings, during the cross-teaching only the NeurIPS provided data was used.

Minor Comments
------------------------
1. Is the base library for applying the training and prediction only Pytorch based or is a Keras version also available?
2. Including a link to the Github code, pypi package etc would be very helpful, this review does not represent a code review and we only focussed on the contents of the paper and the methodology presented.

Major Comments
------------------------
1. Maybe the authors can use recurrence based neural network architecture (ResNet `Kaiming_2016_CVPR`/DenseNet `Huang_2017_CVPR`) for the CNN branch as such networks have superior classification performance and that can potentially boost the performance of the cross teaching network as well. For the future work it would be interesting to see the effect of changing the base network architecture on the performance on the bad segmentation examples.

Quality
----------
The methodical work is of much better quality than the currently existing segmentation solutions in the field of bio-image analysis and it represents the next state of the art.

Clarity
---------
The methods are described very well and the results of good and bad segmentation are presented as separate figures followed by a discussion of the strengths and shortcomings of their method. The paper is within the page limit and is presented in a way that is intuitive and easy to follow.

Conclusions
-----------------
It was a pleasure to read and review this paper, rather than being a delta increment publication this methodology provides a one level up than the current bio-medical image segmentation where iterations of manual relabelling to increment the ground truth training data are current state of the art, therefore this methodology sets the baseline for the next state of the art.

@InProceedings{Huang_2017_CVPR,
author = {Huang, Gao and Liu, Zhuang and van der Maaten, Laurens and Weinberger, Kilian Q.},
title = {Densely Connected Convolutional Networks},
booktitle = {Proceedings of the IEEE Conference on Computer Vision and Pattern Recognition (CVPR)},
month = {July},
year = {2017}
}

@INPROCEEDINGS{Kaiming_2016_CVPR,  author={He, Kaiming and Zhang, Xiangyu and Ren, Shaoqing and Sun, Jian},  booktitle={2016 IEEE Conference on Computer Vision and Pattern Recognition (CVPR)},   title={Deep Residual Learning for Image Recognition},   year={2016},  volume={},  number={},  pages={770-778},  doi={10.1109/CVPR.2016.90}}

---

> ### Author Response · Authors · 2023-02-21
> **Response to Reviewer jKMD**
>
> We thank the reviewer for all the positive comments. We are glad that the manuscript is easy and intuitive to read and understand.
>
> ## Minor Comments:
>
> **[Comment 1]**: ‘Is the base library for applying the training and prediction only Pytorch based or is a Keras version also available?’\
> **Response**: Thank you for the question. MONAI framework is built on top of PyTorch; there is no version compatible with Keras. We have included in the Method section this information about the base library [Pg. 3, Ln. 101].
>
> **[Comment 2]**: ‘Including a link to the Github code, pypi package etc would be very helpful, this review does not represent a code review and we only focused on the contents of the paper and the methodology presented.’\
> **Response**: Thanks for pointing this out. In the abstract, we included the GitHub link [Pg. 1, Ln. 18]. Additional information (e.g. pypi package) can be found in the yml files.
>
> ## Major Comments
>
> **[Comment 1]**: ‘Maybe the authors can use recurrence based neural network architecture (ResNet Kaiming_2016_CVPR/DenseNet Huang_2017_CVPR) for the CNN branch as such networks have superior classification performance and that can potentially boost the performance of the cross teaching network as well. For the future work it would be interesting to see the effect of changing the base network architecture on the performance on the bad segmentation examples.’\
> **Response**: We thank the reviewer for the suggestion. We included a ResUnet [1] in our experiment analysis in Table 4 [Pg 10]. We noticed that the suggested network is performing better than the U-Net, so we thank the reviewer for the suggestion that achieved higher score. We included in the future development the idea of trying different networks to boost the performance of the cross-teaching.
>
> [1] Zhengxin Zhang, Qingjie Liu, and Yunhong Wang. Road extraction by deep residual u-net. IEEE Geoscience and Remote Sensing Letters, 15(5):749–753, 2018.

---

### Official Review · Reviewer_KBrp · 2022-12-19
**CNN and Transformer cross–teaching for multimodal image cell segmentation**

**Rating:** 7
**Confidence:** 4

**Review:**

This paper focuses on the segmentation problem of biological imaging. Traditional algorithms for biological image segmentation must be tailored to the cell type and imaging environment. So semi-supervised or unsupervised multimodal segmentation draws more and more interest.

The authors propose a novel algorithm for transferring and aligning the knowledge between different models with the novel proposed Distance and Neighbour labels. They use UNet and Swin-UNet as the basis of cross-teaching architecture. The proposed method exploits both labelled and unlabeled images and combines the learning paradigms of the two different architectures used.

Minor Concerns:

1. Please cite Diceloss properly.

Major Concerns:

1. In Table 4, baseline and cross-teaching have different numbers of epochs. It would be great if the authors can provide an evaluation with regard to the same number of training epochs.
2. Besides, in Table 4, baseline and cross-teaching using supervised and semi-supervised training. It would be great if authors can provide a baseline with semi-supervised learning.

Questions:

1. (Ablation study) Would representation-based distillation be more effective in transferring knowledge?

This paper is clearly written and with minor grammatical errors which do not affect the reading. The proposed method with Distance and Neighbour labels is novel.

---

> ### Author Response · Authors · 2023-02-21
> **Response to Reviewer KBrp**
>
>
> Thank you very much for your comments that helped us improve this manuscript.
>
> ## Minor Comments
>
> **[Comment 1]**: ‘Please cite Diceloss properly.’\
> **Response**: Revised accordingly [Pg. 4, Ln. 115].
>
> ## Major Comments
>
> **[Comment 1]**: In Table 4, baseline and cross-teaching have different numbers of epochs. It would be great if the authors can provide an evaluation with regard to the same number of training epochs.\
> **Response**: Thanks for the suggestion. We carried out more experiments with our framework and other different networks and we updated Table 4 [Pg. 10] to show the different experiments at epoch 2,000 (the same as the baseline).
>
> **[Comment 2]**: ‘Besides, in Table 4, baseline and cross-teaching using supervised and semi-supervised training. It would be great if authors can provide a baseline with semi-supervised learning.’\
> **Response**: Thank you for this suggestion. Since we had just two weeks to perform all the experiments, we are trying to add this semi-supervised model as a baseline [1], and we hope to insert this experiment for the camera-ready version. For now, we added this idea to future developments.
>
> ## Questions
>
> **[Question 1]**: (Ablation study) Would representation-based distillation be more effective in transferring knowledge?\
> **Response**: Thank you for asking this question. We have found no information about transfer learning applied to a cross-teaching model. However, we found this work [2] in which a student-teacher network was introduced. The authors showed that the student network is performing better than the teacher in transfer learning. In this case, this cannot be applied because we do not have a student-teacher network. In this case, we could investigate if the U-Net or the Transformer is better for transferring learning on another task, but it was already demonstrated in this work that Transformers are better than CNNs for transfer learning when pre-trained on ImageNet and fine-tuned on a biomedical task [3]. However, we added this idea in future developments section [Pg. 11, Ln. 256] once we tried different semi-supervised learning approaches.
>
> [1] Yuchao Wang, Haochen Wang, Yujun Shen, Jingjing Fei, Wei Li, Guoqiang Jin, Liwei Wu, Rui Zhao, and Xinyi Le. Semi-supervised semantic segmentation using unreliable pseudo-labels. In Proceedings of the IEEE/CVF Conference on Computer Vision and Pattern Recognition, pages 4248–4257, 2022.
>
> [2] Junho Yim, Donggyu Joo, Jihoon Bae, and Junmo Kim. A gift from knowledge distillation: Fast optimization, network minimization and transfer learning. In Proceedings of the IEEE conference on computer vision and pattern recognition, pages 4133–4141, 2017.
>
> [3] Mohammad Usman, Tehseen Zia, and Ali Tariq. Analyzing transfer learning of vision transformers for interpreting chest radiography. Journal of digital imaging, 35(6):1445–1462, 2022

---

### Official Review · Reviewer_reKW · 2022-12-24
**Review comment - CNN and Transformer cross–teaching for multimodal image cell segmentation**

**Rating:** 7
**Confidence:** 5

**Review:**

This paper proposed a cross-teaching strategy for multi-modal cell segmentation. By leveraging both labeled and unlabeled data, the proposed model can effectively reduce the annotation burden. This paper is well written and easy to follow. The major concern is the references of the related works, details below.

- Some milestone studies are missing [1][2]. And the reference is incomprehensive and inaccurate. For example, ‘Deep Learning (DL) algorithms have shown encouraging results in fully supervised image segmentation [3]’ In this sentence, authors mentioned fully supervised models, but cited a semi-supervised model. Appropriate references should be carefully added. I suggest citing the SOTA fully supervised nuclei or cell segmentation models here, such as [3][4]. In addition, authors should discuss more about how to reduce the annotation effort [5][6][7].

[1] Pachitariu, M., Stringer, C. Cellpose 2.0: how to train your own model. Nat Methods 19, 1634–1641 (2022). https://doi.org/10.1038/s41592-022-01663-4

[2] Greenwald, N.F., Miller, G., Moen, E. et al. Whole-cell segmentation of tissue images with human-level performance using large-scale data annotation and deep learning. Nat Biotechnol 40, 555–565 (2022). https://doi.org/10.1038/s41587-021-01094-0

[3] Graham S, Vu QD, Raza SEA, Azam A, Tsang YW, Kwak JT, Rajpoot N. Hover-Net: Simultaneous segmentation and classification of nuclei in multi-tissue histology images. Med Image Anal. 2019 Dec;58:101563. doi: 10.1016/j.media.2019.101563.

[4] Zhao B, Chen X, Li Z, Yu Z, Yao S, Yan L, Wang Y, Liu Z, Liang C, Han C. Triple U-net: Hematoxylin-aware nuclei segmentation with progressive dense feature aggregation. Med Image Anal. 2020 Oct;65:101786. doi: 10.1016/j.media.2020.101786.

[5] Zhao T, Yin Z. Weakly Supervised Cell Segmentation by Point Annotation. IEEE Trans Med Imaging. 2021 Oct;40(10):2736-2747. doi: 10.1109/TMI.2020.3046292.

[6] Qu H, Wu P, Huang Q, Yi J, Yan Z, Li K, Riedlinger GM, De S, Zhang S, Metaxas DN. Weakly Supervised Deep Nuclei Segmentation Using Partial Points Annotation in Histopathology Images. IEEE Trans Med Imaging. 2020 Nov;39(11):3655-3666. doi: 10.1109/TMI.2020.3002244.

[7] Han C, Yao H, Zhao B, Li Z, Shi Z, Wu L, Chen X, Qu J, Zhao K, Lan R, Liang C, Pan X, Liu Z. Meta multi-task nuclei segmentation with fewer training samples. Med Image Anal. 2022 Aug;80:102481. doi: 10.1016/j.media.2022.102481.

- ‘so the capacity of the developed model to process data never seen before’, this sentence seems not complete. I guess the authors want to say ‘the capacity of the developed model is limited to process the data it has never seen before’?

- ‘Our solution is based on a cross-teaching between a Convolutional Neural Network (CNN) and a Transformer. This framework takes both labeled and unlabeled images as inputs, and each image passes a CNN and a Transformer respectively to produce the prediction [3]. For the labeled data, the CNN and Transformer are supervised by the ground truth individually [3]. Then the predictions of unlabeled images generated by CNN/Transformer are used to update the parameters of the Transformer/CNN respectively. This framework benefits from the two different learning paradigms, CNNs focus on the local information and transformers model the long range relation, so the cross teaching can help to learn a unified segmenter with these two properties at the same time [3].’
Why authors cite [3] for three times in one paragraph? According to the title of [3], it seems that the idea of this pipeline comes from this paper. If so, authors should clearly point out this and shows the difference between the proposed method and the reference [3].

Authors should carefully proofread the paper, here are some typos:
- trasformer -> transformer
- The U-Net is a 5 layer network -> 5-layer network
- this require a more accurate -> requires

---

> ### Author Response · Authors · 2023-02-21
> **response to Reviewer reKW**
>
> Thank you very much, we are glad that the paper is easy to comprehend. We have read your comments carefully and tried our best to address them one by one.
>
> **[Comment 1]**: Some milestone studies are missing [1][2]. \
> **Response**: Thank you for the suggestion. We added the cited papers in the Introduction section [Pg. 2, Ln. 36] [Pg. 1, Ln. 21].
>
> **[Comment 2]**: And the reference is incomprehensive and inaccurate. For example, ‘Deep Learning (DL) algorithms have shown encouraging results in fully supervised image segmentation [3]’ In this sentence, authors mentioned fully supervised models, but cited a semi-supervised model. Appropriate references should be carefully added. \
> **Response**: Thank you for pointing out the citation error. We changed the citation accordingly.
>
> **[Comment 3]**: I suggest citing the SOTA fully supervised nuclei or cell segmentation models here, such as [3][4]. \
> **Response**: Thank you for the suggestion. We added the cited papers in the Introduction section [Pg. 2, Ln. 31].
>
> **[Comment 4]**: In addition, authors should discuss more about how to reduce the annotation effort [5][6][7].\
> **Response**: We added more information about the annotation effort, and we added the suggested paperers [Pg. 2, Ln. 40].
>
> **[Comment 5]**: ‘so the capacity of the developed model to process data never seen before’, this sentence seems not complete. I guess the authors want to say ‘the capacity of the developed model is limited to process the data it has never seen before’?\
> **Response**: Thank you for pointing out the problem in the quoted sentence. We changed the paragraph to add the suggested paper and better develop the annotation problem [Pg. 2, Ln. 30].
>
> **[Comment 6]**: ‘Our solution is based on a cross-teaching between a Convolutional Neural Network (CNN) and a Transformer. This framework takes both labeled and unlabeled images as inputs, and each image passes a CNN and a Transformer respectively to produce the prediction [3]. For the labeled data, the CNN and Transformer are supervised by the ground truth individually [3]. Then the predictions of unlabeled images generated by CNN/Transformer are used to update the parameters of the Transformer/CNN respectively. This framework benefits from the two different learning paradigms, CNNs focus on the local information and transformers model the long range relation, so the cross teaching can help to learn a unified segmenter with these two properties at the same time [3].’ Why authors cite [3] for three times in one paragraph? According to the title of [3], it seems that the idea of this pipeline comes from this paper. If so, authors should clearly point out this and shows the difference between the proposed method and the reference [3].\
> **Response**: Thank you to have pointed out these citation errors. We have modified the paragraph as you suggested. We changed Section 2 [Pg. 2, Ln. 68] in which we pointed out the differences between our method and the reference method.
>
> **[Comment 7]**:  Authors should carefully proofread the paper, here are some typos:
> trasformer -> transformer
> The U-Net is a 5 layer network -> 5-layer network
> this require a more accurate -> requires\
> **Response**: Thank you to have pointed out the typos. We have corrected at least the ones indicated by the reviewer.

---

### Official Review · Reviewer_4dc3 · 2023-01-09
**Review comment of 'CNN and Transformer cross-teaching for multimodal image cell segmentation'**

**Rating:** 7
**Confidence:** 5

**Review:**

This paper provided a cross teaching frame for cell segmentation. With cross teaching the solution proposed in this paper outperformed the official baseline.
There are several expressions could be updated, which make this paper more rigorous and clearly reading:
1. 'the distance from the center of the cell' and 'the neighbor distance between cells' are not 'class'. They are properties of a pixel rather than a semantic class.
2. The words could be more brief. For example, section 2 (method) could be more clear with implementaion details moved to the experiment section, and the decription for U-Net structure in section 2.2 could be shorten to a cite in a sentense.
3. The paper didn't point out the difference between the proposed framework and the cited works.

---

> ### Author Response · Authors · 2023-02-21
> **Response to Reviewer 4dc3**
>
> Thank you very much for your previous comments that helped us improve this manuscript.
>
> **[Comment 1]**: 'the distance from the center of the cell' and 'the neighbor distance between cells' are not 'class'. They are properties of a pixel rather than a semantic class.\
> **Response**: Thank you for pointing out this problem with the labels. We changed the terminology for the distance and neighbour labels from 'class' to 'maps'.
>
> **[Comment 2]**: The words could be more brief. For example, section 2 (method) could be more clear with implementaion details moved to the experiment section, and the decription for U-Net structure in section 2.2 could be shorten to a cite in a sentence.\
> **Response**: Unfortunatelly we are required from the organizers to maintain the current manuscript organization
>
> **[Comment 3]**: The paper didn't point out the difference between the proposed framework and the cited works.\
> **Response**: Agree. We have, accordingly, modified Section [Pg. 2, Ln. 68] to emphasize this point. In Section 3.2.2 of the Experiment section, we pointed out that the method provided by the organizer is considered the baseline for our method and we have added more information for the training procedure and results Section 2.1.

---

### Official Review · Reviewer_fLzy · 2023-01-12
**A cross-teaching framework for weakly supervised cell segmentation**

**Rating:** 6
**Confidence:** 4

**Review:**

The paper proposes a framework that utilizes both CNN and Transformer for weakly supervised cell segmentation. CNN model is supposed to provide local information, Transformer is supposed to provide long range relation.

The good thing is that the paper proposes a complete solution, and it utilizes both labeled data and unlabeled data. However, the reviewer still have the following concerns:

1. The paper proposes a method for weakly supervised. However, the motivation and the efficiency of the method is not discussed. There are many questions about the proposed method have not been answered in the paper. For example: Why the predictions of CNN and Transformer are good pseudo labels for the other model? How to deal with error propagation issue? With the proposed training scheme, it is very likely that both CNN and Transformer make the same errors in the final predication.

2.  One of the challenges in cell segmentation is to segment cells correctly in crowded cases. Local information is always crucial to separate cells correctly. However, in the proposed method, how to deal with this challenge is not mentioned. It uses Transformer to provide ``long range relation''. What kind of long range relation the transformer provides? How does it help to segment cells?

3. Some details are missing. How to obtain the final instance segmentation results with two different outputs?

---

> ### Author Response · Authors · 2023-02-21
> **Response to Reviewer fLzy**
>
> Thank you very much for agreeing with us on the intention of this manuscript. Thank you for your comments. We have gone through your comments carefully and tried our best to address them one by one. We hope the manuscript has been improved accordingly.
>
> **[Comment 1]**: The paper proposes a method for weakly supervised. However, the motivation and the efficiency of the method is not discussed. There are many questions about the proposed method have not been answered in the paper. For example: Why the predictions of CNN and Transformer are good pseudo labels for the other model? How to deal with error propagation issue? With the proposed training scheme, it is very likely that both CNN and Transformer make the same errors in the final predication.\
> **Response**: Thank you for this question. We added in the Introduction a more detailed explanation about the choice of using the pseudo-labels of the other model [Pg. 2, Ln. 52] and also in the Method section [Pg. 2, Ln. 74]. In Section 4.1 we have conducted an ablation study for the unlabelled images to point out the difference between the model before and after adding the unlabelled images. The two networks' predictions are different, and the two networks learn with two different rhythms (the Swin + U-Net is faster). In Section 4.5 we added a part about error propagation using pseudo-labels. As presented in [1], this learning paradigm is good to deal with pseudo-label errors but is not the best. Hence, we would like to perform further analysis in the future.
>
> **[Comment 2]**: One of the challenges in cell segmentation is to segment cells correctly in crowded cases. Local information is always crucial to separate cells correctly. However, in the proposed method, how to deal with this challenge is not mentioned. It uses Transformer to provide ``long range relation''. What kind of long range relation the transformer provides? How does it help to segment cells?\
> **Response**: Thank you to point this out, we have added an explanation of the “long-range relation” in the Introduction section and added some papers that can clarify this concept even further [Pg. 2, Ln. 52].
>
> **[Comment 3]**: Some details are missing. How to obtain the final instance segmentation results with two different outputs?\
> **Response**: Thank you for the question. For the post-processing, we used the one provided by the organizers to speed up the inference time, which is part of the evaluation. Future development would include more sophisticated post-processing with the five outputs of the network.
>
> [1] Ma, C., Pan, X., Ye, Q., Tang, F., Dong, W., & Xu, C. (2023). CrossRectify: Leveraging disagreement for semi-supervised object detection. Pattern Recognition, 137, 109280.

---

### Decision · Program_Chairs · 2023-01-19

Accept